# Positive Prognostic Overall Survival Impacts of Methylated *TGFB2* and *MGMT* in Adult Glioblastoma Patients

**DOI:** 10.3390/cancers17071122

**Published:** 2025-03-27

**Authors:** Sanjive Qazi, Michael Potts, Scott Myers, Stephen Richardson, Vuong Trieu

**Affiliations:** 1Oncotelic Therapeutics, 29397 Agoura Road, Suite 107, Agoura Hills, CA 91301, USA; michael.potts@oncotelic.com (M.P.); scott.myers@oncotelic.com (S.M.); stephen.richardson@oncotelic.com (S.R.); vtrieu@oncotelic.com (V.T.); 2Westmorland Campus, Kendal College, Market Place, Kendal, Cumbria LA9 4TN, UK

**Keywords:** biomarker, brain neoplasms, gliomas, prognosis, transforming growth factors, tumor microenvironment, immunotherapy, T-cell

## Abstract

Glioblastoma (GBM) is a highly aggressive brain tumor in adults. It is well established that O-6-methylguanine-DNA methyltransferase (*MGMT*) gene methylation predicts overall survival (OS) benefits for patients treated with standard temozolomide and radiotherapy regimens. Transforming growth factor beta (TGFB) is a multifunctional family of cytokines involved in cellular processes and regulating growth factors. Our novel finding suggests that high *TGFB2* gene methylation levels are associated with an improved overall survival risk compared to *MGMT* and *TGFB1* methylation, controlling for age and sex. We identified several genes and pathways linked to *TGFB2* methylation, involving immune system mechanisms implicated in T-cell activation, antigen processing, and Toll-like receptor pathways to improve survival outcomes in GBM patients. In particular, MALT1 (mucosa-associated lymphoid tissue lymphoma translocation protein 1) mRNA negatively impacted survival rates, providing a potential target for therapies.

## 1. Introduction

Glioblastoma (GBM) is the most aggressive and common primary malignant brain tumor in adults; it is characterized by its poor prognosis and high mortality rate. The incidence of glioblastoma accounts for approximately 45.6% of all primary brain tumors in adults. The annual incidence of GBM is reported to be between two and three new cases per 100,000 population [1]. According to the WHO 2021 classification schema, GBM is currently defined as diffuse, grade 4, astrocytoma lacking isocitrate dehydrogenase 1 gene (*IDH)* and histone mutations that have concomitant +7/−10, epidermal growth factor receptor gene (*EGFR)* amplification, telomerase reverse transcriptase (*TERT)* promoter mutations, mitoses, necrosis, or microvascular proliferation [1,2,3]. Newly diagnosed patients with GBM exhibit a median survival time of approximately 15–18 months and a 5-year survival rate below 10% [1]. Epigenetic modifications in GBM tumors, including *MGMT* methylation, were found to predict the survival benefit of being treated with standard temozolomide and radiotherapy regimens [2,3,4,5,6].

Bioinformatic analysis of transforming growth factor beta 2 (TGFB2) mRNA has revealed its significant role as a negative prognostic marker in low-grade glioma, where mRNA levels of TGFB2 are upregulated in low-grade glioma (LGG) tumors and correlated with poorer overall survival outcomes. Additionally, an independent prognostic study of the impact of high TGFB2 mRNA levels and isocitrate dehydrogenase wildtype (*IDHwt*) mutation status suggested TGFB2’s potential utility in risk stratification for clinical trials. Moreover, TGFB2, in conjunction with other markers such as interferon regulatory factor 5 (IRF5) and CD276/B7-H3, may be targeted for combination therapies [7]. TGFB2 may have a negative impact on survival in glioblastoma (GBM) patients, as suggested by increased survival duration at increased IL-1beta mRNA levels or low TGFB2 levels [8]. High levels of TGFB2 in glioma cells contributed to the malignant phenotype of glioblastoma, according to observations showing that antisense oligonucleotides, ISTH0047, inhibit TGFB2 expression, resulting in a reduction in glioma cell migration and invasiveness, as well as a decrease in the phosphorylation of the downstream target SMAD2 in various glioma cell lines [9]. Furthermore, systemic administration of the antisense oligonucleotide targeting TGFB2 in glioma-bearing mice led to prolonged survival and increased immune cell infiltration within tumors. This indicates that TGFB2 plays a significant role in the modulation of GBM progression and immune response. This observation was further supported in a clinical study that employed OT101, a TGFB2-specific synthetic phosphorothioate antisense oligodeoxynucleotide (S-ODN). This led to considerable reductions in tumor size among patients with GBM and anaplastic astrocytoma after 4 to 11 cycles of OT101. A total of 25% of 77 patients experienced lasting objective responses, with six individuals attaining a complete 100% reduction in the three-dimensional volume of their targeted lesions. [10].

Given the prognostic impact of *MGMT* gene methylation and TGFB2 mRNA expression levels in pediatric diffuse midline gliomas [11,12] and low-grade glioma patients [7], we extended this study using a similar bioinformatic-led approach to elucidate the prognostic impact of *TGFB2* and *MGMT* gene methylation and to characterize their effect on overall survival (OS) in adult GBM patients. Central to our method was the implementation of a multivariate Cox proportional hazards model to directly compare hazard ratio (HR) calculations for *TGFB1/2/3* and *MGMT* methylation on OS, considering male versus female, age at diagnosis, and age interactions with *TGFB2* gene methylation and sex variables. To identify prognostic gene markers, we screened the mRNA expression of 14364 genes curated across 1272 Reactome pathways to identify consistently affected pathways, comparing young, old, and all GBM patients enriched with the expression of genes negatively correlated with *TGFB2* methylation. Our results suggest that *TGFB2* gene methylation serves as a strong prognostic marker when considering age and sex, preferentially benefiting young adult male patients.

## 2. Materials and Methods

### 2.1. AI-Augmented Summaries of PubMed Abstracts

PubMed searches using the keywords “GBM AND MGMT AND methylation” retrieved 841 abstracts; “Glioblastoma AND Prognosis AND Macrophage” (490 abstracts); “Glioblastoma AND Prognosis AND single-cell RNA-seq” (742 abstracts); and “Glioblastoma AND Prognosis AND TGF-beta” (654 abstracts) (https://pubmed.ncbi.nlm.nih.gov/ accessed 25 October 2024) were downloaded as text documents for processing using the Oncotelic Chatbot technologies. Each abstract was then indexed into our Quadrant database (aided using puppeteer 19.11.1), embedded, and transformed (langchain-openai 0.2.3, openai 1.52.0) into a vector of numbers capturing semantic similarity between text elements (tokens). The embedding transforms all abstracts to the same vector “embedding” space and has been trained to minimize the distance (in vector space) between any pair of abstracts to the extent that they are semantically similar. Using an agglomerative clustering algorithm (hdbscan 0.8.39) to group the vectors, we automatically labeled these clusters using the question-answering model to identify any similarity between the abstracts corresponding to each cluster’s vectors. In the question-answering session, the user query was transformed into an embedding vector, and then an appropriate similarity measure (e.g., cosine distance) was used to identify the embedded abstract vectors closest to the embedded query vector: the abstracts corresponding to these matching vectors were then fed to the question-answering model (in the form of context), along with the query, to arrive at an answer to the query.

The user interface is powered by the React framework, an open-source and flexible language for developing powerful front-end interfaces (https://react.dev/ accessed 25 March 2024). In addition, we used the @mui/material (https://mui.com/ accessed on 25 March 2024)) libraries for the interface’s design aspects and aimed to follow closely the material design guidelines. Serving the front end was Node.Js (https://nodejs.org/en Accessed 25 March 2024). The Node.js libraries included in the project were @adobe/pdfservices-node-sdk 3.4.2@aws-sdk/client-s3 3.412.0, @langchain/community 0.2.5, @material-ui/core 4.12.4, @mui/base 5.0.0-beta.18, @mui/icons-material 5.11.16, @mui/material 5.15.20, @mui/styled-engine-sc 5.12.0, @mui/x-date-pickers 6.15.0, @qdrant/js-client-rest 1.4.0, carrot2 0.0.1, pdf2img 0.5.0, pdfjs-dist 4.5.136, puppeteer 19.11.1, react 18.2.0, sequelize 6.31.1, and zod 3.22.4.

The AI-augmented interrogation of PubMed enabled the rapid identification of key primary publications pertinent to this research project.

### 2.2. Demethylation Impact on OS for Glioblastoma Patients (GBM) of the Target Genes

We analyzed clinical metadata and methylation beta values obtained from normalized values merged between the hm27 and hm450 platforms for 292 patients who were diagnosed with GBM according to the WHO2021 definition, after removing five patients with IDH mutations from the 2018 dataset (https://www.cbioportal.org/study/summary?id=gbm_tcga_pan_can_atlas_2018; data file: “data_methylation_hm27_hm450_merged.txt” accessed on 22 February 2024, or can be accessed via https://www.cbioportal.org/datasets and then by clicking onto Glioblastoma Multiforme (TCGA, PanCancer Atlas) accessed on 22 February 2024). We examined the overall survival (OS) impact on patients with lower methylation levels of the target genes *TGFB1/2/3* and *MGMT*, using the 25th percentile cut-off for each gene, and compared it to that of the remaining patients.

### 2.3. Comparing mRNA Expression in GBM Tumors and Normal Brain Samples

Expression levels of the genes were provided as log2 transformed TPM (transcripts per million, calculated using the RSEM algorithm) and obtained as RNAseq data files (https://toil-xena-hub.s3.us-east-1.amazonaws.com/download/TcgaTargetGtex_rsem_gene_tpm.gz; complete metadata, accessed on 25 July 2023) from the UCSC Xena web platform (https://xenabrowser.net/datapages/ accessed on 25 July 2023) [13]. In this analysis, the mRNA expression levels (transcripts per million: TPM) for 99 GBM patients were expressed as fold change values relative to 1141 brain samples, facilitated by the standardized recalculated gene and transcript expression dataset for all tumor and normal tissue samples [14].

### 2.4. Hazard Ratio Comparisons for GBM Patients to Determine the Independent Effect of High Levels of TGFB1/2/3, MGMT Methylation Levels, Age, Sex

Multivariate analyses utilized the Cox proportional hazards model to assess the individual effects of *TGFB2* methylation on OS (N = 101, 73 death events). This analysis controlled for age at diagnosis and the interaction between *TGFB2* and age, and between *TGFB2* and sex. Briefly, the model included (i) methylated *TGFB2* levels (upper quartile cut-off); (ii) methylated *TGFB1* levels (upper quartile cut-off); (iii) methylated *TGFB3* levels (upper quartile cut-off); (iv) methylated *MGMT* levels (upper quartile cut-off); (v) age at diagnosis as a linear covariate; (vi) *TGFB2* × age interaction; and (vii) age × sex interaction. All calculations were performed using the R platform (survival_3.2-13 ran in R version 4.1.2). Hazard ratios were estimated using the exponentiated regression coefficients and were visualized using forest plots (survival_3.2-13 and survminer_0.4.9 ran in R version 4.1.2 (1 November 2021)).

### 2.5. Geneset Enrichment Analysis (GSEA) to Identify Negatively Correlated Pathways to TGB2 Methylation Levels

The present study analyzed *TGFB1/2* methylation levels and their Spearman correlation coefficient against cohorts of patients stratified according to age and sex. Normalized enrichment scores (NESs) for 1272 Reactome pathways across 14364 evaluable genes were subsequently computed and ranked (implemented using the fgsea version 1.20.0 package in R). The permutation *p*-values for normalized enrichment scores were determined, identifying pathways significantly correlated with *TGFB1/2* methylation. The permutation *p*-values for the normalized enrichment scores were determined, identifying significantly correlated pathways with the cohorts of GBM patients. We used a two-way hierarchical clustering technique to organize the NES scores for the most affected pathways (*p* < 0.001) across all the cohorts of GBM patients based on age and sex. The NES profiles were grouped using the average distance metric (default Euclidean distance), identifying columns of GBM cohorts and rows of Reactome pathways (heatmap.2 function in the R package gplots_3.1.1).

### 2.6. mRNA Expression Validation of Marker Genes Identified in Bulk mRNA Using Single-Cell Determinations in GBM Samples

The aim is to validate the expression of the 23 genes; these genes were upregulated in tumor tissues and predicted the prognostic impact on OS from the bulk mRNA obtained from the TCGA dataset, using an independent dataset utilizing RNA sequencing quantification from single cells obtained from 20 adult and 8 pediatric *IDH*-wildtype glioblastoma patients (GEO: GSE131928, https://singlecell.broadinstitute.org/single_cell/study/SCP393/single-cell-rna-seq-of-adult-and-pediatric-glioblastoma, accessed on 27 February 2025). The authors used a t-distributed stochastic neighbor embedding (tSNE) plot of all the single cells and visualized it in 2 dimensions to capture clusters of cell types. The cells were stratified according to cell type (macrophage (754 cells), malignant (6863 cells), oligodendrocytes (219 cells), and T-cells (94 cells) and GBM type (adult 5742 cells and pediatric 2188 cells) [15].

## 3. Results

### 3.1. Positive Prognostic Impact of High TGFB2 and MGMT Gene Methylation Levels on Overall Survival (OS) in Adult GBM Patients

Alone, age at diagnosis and sex impacted OS in GBM patients (Appendix A). These patients were stratified into old and remaining patients (the cut-off at the upper quartile of 69 years was the age at which there was a significant separation of the survival curves) and male versus female patients. The median OS time for 78 old patients was 7.63 (95% CI: 4.87–10.85, death events = 61) months, which was significantly (Log-rank Chi-Square = 38.64, *p* < 0.001) shorter than the 214 remaining patients (median OS = 16 (95% CI: 14.93–17.65) months, death events = 146) (Appendix A). The old and remaining patients predicted a parallel shift in the survival times with a difference of 8.37 months in the median OS times (mOS). Female patients (N = 121, median OS = 15.85 (95% CI: 12.49–19.3) months, death events = 82) predicted a significant (Log-rank Chi-Square = 38.64, *p* < 0.001) improvement in the survival outcome compared to the male patients (N = 171, median OS = 13 (95% CI: 12.23–14.93) months, death events = 125; mOS difference of 2.85 months) in this cohort of GBM patients (Appendix A). Examination of the OS curves showed that male and female patients followed similar trajectories until 20 months, after which the male patients diverged, contributing to the significant difference in survival times (*p* < 0.001).

On comparing GBM patients with high versus low levels of gene methylation for *TGFB1/2/3* and *MGMT,* positive prognostic impacts were observed for patients with high levels of *TGFB2* and *MGMT* methylation (Appendix A). GBM patients (N = 73) with high levels of *TGFB2* methylation (*TGFB2^highMe^*) predicted a median OS time of 14 (95% CI: 12.4–23.1, death events = 50) months, which was similar to the median OS time of 14.4 (95% CI: 12.5–15.3, death events = 157) months for 219 patients with lower methylation levels (*TGFB2^lowMe^*). At later time points (>20 months), the high and low methylation groups significantly diverged in this Kaplan–Meier analysis with a better prognosis for the *TGFB2^highMe^* group of patients (Log-rank Chi-Square = 3.971, *p* = 0.046) (Appendix A). This trajectory was similar to the survival times observed when comparing male and female patients. The median OS time for 73 patients with high levels of *MGMT* methylation (*MGMT^highMe^*) was 16.8 (95% CI: 15.1–21.7, death events = 45) months and was significantly longer than the median OS time for the remaining patients (N = 219, 12.7 (95% CI: 11.5–14.9, death events = 162) months; Log-rank Chi-Square = 7.729, *p* = 0.005; mOS difference of 4.1 months) (Appendix A). GBM patients (N = 73) in the *TGFB1^highMe^* group predicted an OS time of 17.2 (95% CI: 12.2–23.1, death events = 42) months, which was longer than the remaining patients but did not reach statistical significance (N = 219; 13.6 (95% CI: 12.5–15, death events = 165) months; Log-rank Chi-Square = 3.804, *p* = 0.051) (Appendix A). Similar outcomes were observed for the *TGFB3^highMe^* group of GBM patients, where the median OS time for the high methylation group was 15.0 (N = 73; 95% CI: 11.8–21.7, death events = 53) months compared to 14.0 (95% CI: 12.5–15.3, Events = 154) months for the 219 remaining patients (Log-rank Chi-Square = 2.22, *p*-value = 0.136) (Appendix A).

The GBM patients were interrogated using the TCGA dataset to correlate OS with methylation status stratified according to both sex and age at diagnosis to investigate the prognostic impact of *TGFB2* (Appendix A)*, TGFB1* (Appendix A), *MGMT* (Appendix A), and *TGFB3* (Appendix A) methylation in these patients. The impact of methylation on this gene complex is complicated by the overwhelmingly strong impact of age and sex on survival (Appendix A). Additionally, unlike *MGMT* methylation, which has a consistent prognostic effect across the age range, *TGFB2* methylation predicts a flip from being positively prognostic to being negatively prognostic, going from young to old patients (Appendix A). We observed prolonged survival times in the young male and female GBM patients (Appendix A). All the subgroups of old GBM patients stratified by sex and gene methylation levels (high/low for *TGFB1/2/3* and *MGMT*) experienced mOS times ranging from 7 to 12.7 months. In contrast, the subgroups of young GBM patients experienced mOS times ranging from 14 to 37 months (Appendix A). Young females experienced mOS times ranging from 21 to 37 months across all high and low levels of *TGFB1/2/3* and *MGMT* gene methylations. The young males experienced mOS times ranging from 14 to 23 months (Appendix A). In the examination of the *TGFB2* methylation impact on OS, the younger males predicted improved survival outcomes, where 58 young male patients with low levels of *TGFB2* methylation experienced an mOS of 14.1 (95% CI: 12.2–17.6, death events = 43) months; this survival outcome was significantly improved at high levels of *TGFB2* methylation to 23.1 (N = 22; 95% CI: 12.6–NA, death events = 12) months; *p* = 0.05 (Appendix A). The examination of the impact of *TGFB3* methylation showed borderline increases in mOS for both males and females; we observed an mOS of 14 months for young male patients with low levels of *TGFB3* methylation and a prolonged mOS of 18.1 months at high levels of *TGFB3* methylation (*p* = 0.06) (Appendix A). The young female patients predicted an mOS of 20.8 months at low levels of *TGFB3* methylation and 36.9 months at increased *TGFB3* methylation levels (*p* = 0.06) (Appendix A).

We next investigated the combined prognostic impact of *MGMT* and *TGFB1/2/3* methylations in a young cohort of GBM patients to evaluate whether *MGMT* and *TGFB1/2/3* methylations further prolonged survival times (upper quartile percentile cut-off for high methylation levels and age at diagnosis; the median cut-off for young patients (<60 years) was used for both male and female patients to obtain sufficient sample sizes for statistical comparisons) (Figure 1). Kaplan–Meier analysis was performed for four combinations of *TGFB1/2/3* and *MGMT* methylations. A subset of 15 patients with high levels of *TGFB2* and *MGMT* methylation experienced prolonged survival times (mOS = 60 months) compared to 75 patients with low levels of both *TGFB2* and *MGMT* methylation (mOS = 15 months; Adj. *p* = 0.032). There was a significant trend of increasing mOS across all four subsets of GBM patients (*p* = 0.019; *TGFB2^lowMe^*/*MGMT^highMe^* predicted mOS of 21.1 months, patients with *TGFB2^highMe^*/*MGMT^lowMe^* predicted mOS of 21.2 months) (Figure 1A). The patients with *TGFB1* and *MGMT* methylation showed an increasing trend of mOS related to increasing methylation levels (*p* = 0.051): the mOS for *TGFB1^lowMe^*/*MGMT^lowMe^* group of patients was 14.5 months; the mOS for *TGFB1^lowMe^*/*MGMT^highMe^* group of patients was 17.9 months; the mOS for *TGFB1^highMe^*/*MGMT^lowMe^* group of patients was 18.6 months; and the mOS for *TGFB1^highMe^*/*MGMT^highMe^* group of patients was 36.9 months. The increase in the mOS difference between patients with high levels of *TGFB1* and *MGMT* methylation compared to those with low levels of *TGFB1* and *MGMT* methylation was borderline significant (14.5 versus 36.9 months, respectively; Adj. *p* = 0.051) (Figure 1B). The patients with *TGFB3* and *MGMT* methylation showed a significant increasing trend of mOS related to increasing methylation levels (*p* = 0.015): the mOS for *TGFB3^lowMe^*/*MGMT^lowMe^* group of patients was 14.5 months; the mOS for *TGFB3^lowMe^*/*MGMT^highMe^* group of patients was 21.1 months; the mOS for *TGFB3^highMe^*/*MGMT^lowMe^* group of patients was 22.5 months; and the mOS for *TGFB3^highMe^*/*MGMT^highMe^* group of patients was 36.9 (95% CI: 17.8–NA, death events = 7) months. The increase in the mOS difference between the patients with high levels of *TGFB3* and *MGMT* methylation compared to those with low levels of *TGFB3* and *MGMT* methylation was borderline significant (14.5 versus 36.9 months, respectively; Adj. *p* = 0.061) (Figure 1C).

### 3.2. Independent Prognostic Impacts of TGFB1/2/3, MGMT Gene Methylation, Controlling for Age, Sex, and Determining the Impact of Age Interactions on OS in GBM Patients

We tested for cross-correlations of variables that may impact OS, thereby confounding the impact of methylated genes in our analysis. In all the GBM patients, *TGFB2* methylation was positively correlated with *TGFB1* (r = 0.35, *p* < 0.0001), *TGFB3* (r = 0.22, *p* = 0.0002), and *MGMT* (r = 0.21, *p* = 0.0003) methylation and negatively correlated with the age at diagnosis (r = −0.3, *p* < 0.0001) (Appendix A). Correlation of variables in the male patients showed that *TGFB2* methylation was positively correlated with *TGFB1* (r = 0.4, *p* < 0.0001), *TGFB3* (r = 0.28, *p* = 0.0002), and *MGMT* (r = 0.24, *p* = 0.0016) methylation and negatively correlated with the age at diagnosis (r = −0.29, *p* = 0.0001) (Appendix A). Female patients exhibited a weaker correlation than male patients, where *TGFB2* methylation was only correlated with *TGFB1* methylation (r = 0.2, *p* = 0.028) and age at diagnosis (r = −0.32, *p* = 0.0003) (Appendix A). Due to the complex nature of *TGFB2* methylation in subsets of GBM patients and survival, we used the multivariate model to further substantiate the impact of TGFB2 as an independent variable considering the correlation with other variables, including age, sex, and methylation of *TGFB1/3* and *MGMT* methylation. We employed the multivariate Cox proportional hazards model to investigate the independent impact of *TGFB2* methylation on OS times. This technique first constructs a baseline survival curve that includes all patients in the dataset, then utilizes the fitted parameters to shift the baseline curves using values fixed for the control variables and varying a single variable of interest, thereby evaluating the independent impact of the varied parameter. The multivariate model showed a highly significant effect of age at diagnosis (*p* < 0.001), age interactions with *TGFB2* methylation (*p* = 0.001), and age with sex (*p* = 0.014) (Figure 2). The multivariate analysis of *TGFB2* versus *MGMT* gene methylations reveals this multiplicative impact of age as a significant interaction term.

### 3.3. Age-Independent Identification of Reactome Pathways Enriched with mRNA Expression for Genes Negatively Correlated with TGFB2 Gene Methylation

The multivariate Cox regression models showed the improved OS impact of high levels of *TGFB2* methylation for both the male and female patients, with minimal sexual dimorphism in the young GBM patients; we examined Reactome pathways that showed consistent enrichment of negatively correlated mRNA expression for genes across old, young, and all GBM patients (Figure 3, Appendix A). Sixty-four Reactome pathways showed significant enrichment of negatively correlated genes across all subsets of patients based on age as a stratifying factor. These included the Reactome pathways that collectively represented the complex signaling cascade that occurs following TCR engagement, leading to T-cell activation, differentiation, and effector functions (phosphorylation of CD3 and TCR zeta chains12; translocation of ZAP-70 to immunological synapse; generation of second messenger molecules; downstream TCR signaling; CD28-dependent Vav1 pathway; co-stimulation by the CD28 family; PD-1 signaling; Interleukin-2 family signaling); Interleukin signaling mechanisms (“Interleukin-12 family signaling”, “Interleukin-2 family signaling”, “Interleukin-4 and Interleukin-13 signaling”, “Interleukin-10 signaling”, and “Interleukin receptor SHC signaling”, “Interleukin-3, Interleukin-5 and GM-CSF signaling”); Interferon pathways (“Interferon alpha/beta signaling”, “Interferon gamma signaling”, and); antigens interactions (“Antigen Processing-Cross presentation” and “Immunoregulatory interactions between a Lymphoid and a non-Lymphoid cell”); and Toll-like receptor cascades (“Toll Like Receptor TLR6:TLR2 Cascade”, “Toll Like Receptor 2 (TLR2) Cascade”, “Toll Like Receptor TLR1:TLR2 Cascade”, “Toll Like Receptor 3 (TLR3) Cascade”, “MyD88-independent TLR4 cascade”, “TRIF(TICAM1)-mediated TLR4 signaling”, “Toll Like Receptor 7/8 (TLR7/8) Cascade”, “Toll Like Receptor 9 (TLR9) Cascade”, “ IRAK4 deficiency (TLR2/4)”, “MyD88 deficiency (TLR2/4”), “Diseases associated with the TLR signaling cascade”, and “Regulation of TLR by endogenous ligand”) (Appendix A).

### 3.4. Genes Negatively Correlated with TGFB2 Methylation Predicted Significant mRNA Increases in Tumor Tissue with Positive Prognostic Impact on OS in GBM Patients

Sixty-four Reactome pathways were enriched in the mRNA expression of genes negatively correlated with *TGFB2* methylation. The pathways negatively correlated with *TGFB2* methylation were represented by 1161 genes, of which 275 predicted significant negative correlations with *TGFB2* methylation (*p* < 0.05, FDR = 0.11). Examination of the levels of the mRNA expression of these 275 genes (Appendix A) showed that 169 genes were significantly upregulated in tumor tissues (*p* < 0.0001). The expression of these genes was screened for prognostic impact using a multivariate Cox proportion hazards model measuring the effects of the mRNA expression of each of the genes and considering the impact of *TGFB1/2/3* and *MGMT* methylation, controlling for age, sex, and age interactions; it yielded 20 genes that showed an OS impact for the gene and *MGMT* methylation upregulated in tumor tissues (Appendix A) in addition to genes representing CD3D (T-cell marker), CD3E (T-cell marker), and CD86 (M1-like macrophage marker) (Figure 4A). To validate the bulk RNA-seq from the TCGA study, we analyzed the expression of the 23 genes in an independent dataset utilizing RNA sequencing quantification from single cells obtained from 20 adult and 8 pediatric *IDH*-wildtype glioblastoma patients. In adult patients, PARP9 and IFI35 predicted high levels of expression in macrophages; TRIM22 and MALT1 were highly expressed in macrophages and T-cells; TRIM5 and HIF1A were highly expressed in macrophages and malignant cells; IRAK4 was highly expressed in macrophages, malignant cells, and T-cells (Figure 4B,C and Appendix A). In the multivariate models, increasing age and male patients predicted worse survival outcomes. This prompted us to identify the favorable prognostic impact of genes identified in the TCGA and single-cell RNA-seq experiments. We utilized fitted parameters from the multivariate Cox proportional hazards model to generate predicted survival curves for increasing HIF1A, TRIM22, IRAK4, and PARP9 mRNA levels for these genes expressed in macrophages (HIF1A: 89.26% of macrophage cells, 4.82 Log2 TPM; TRIM22: 80.5%, 3.22 Log2 TPM; IRAK4 92.18%, 1.16 Log2 TPM; PARP9: 50.8%, 1.45 Log2 TPM). These genes showed significant positive prognostic impact independently of *TGFB2* and *MGMT* methylation (Appendix A).

Next, we investigated the methylation load (sum of *TGFB1/2/3* and *MGMT* beta values) and the mRNA expression values of the genes highly expressed in macrophages. An examination of 101 GBM patients for whom clinical, mRNA expression, and methylation data were available showed significantly improved survival times for the patients with a high methylation load (Figure 5A; mOS = 21.3 versus 14.5 months for high versus low methylation load, respectively). Kaplan–Meier analysis of the GBM patients stratified by methylation load and mRNA expression (Figure 5B–D) showed significant improvements in survival times for the patients expressing high levels of HIF1A, TRIM22, and PARP9 mRNA and a high methylation load, confirming the results obtained from the Cox proportional hazards models.

The multivariate model that included TRIM5 mRNA predicted significant effects for TRIM5 mRNA expression, which was found to be 0.607 (95% CI range: 0.413–0.892; *p* = 0.011). Additionally, the hazard ratio (HR) for increasing age at diagnosis was 1.059 (95% CI range: 1.023–1.096; *p* = 0.001), while the interaction term for *TGFB2* by age was 1.057 (95% CI range: 0.992–1.127; *p* = 0.085), and for the age by sex, it was 0.95 (95% CI range: 0.903–0.999; *p* = 0.046). Regarding the other methylation impacts, the HR for *TGFB2^highMe^* was 0.016 (95% CI range: 0–0.919; *p* = 0.045), *TGFB1^highMe^* was 0.731 (95% CI range: 0.333–1.605; *p* = 0.435), and *TGFB3^highMe^* was 0.629 (95% CI range: 0.309–1.279; *p* = 0.2), and for *MGMT^highMe^*, the HR was 0.403 (95% CI range: 0.217–0.749; *p* = 0.004) (Figure 6).

MALT1 (Reactome pathway: “CLEC7A (Dectin-1”) signaling) mRNA expression was the only gene product with a negative prognostic impact on OS in the GBM patients (Figure 7). The HR for MALT1 mRNA expression was 1.997 (95% CI range: 1.1–3.625; *p* = 0.023). With increasing age, the HR was 1.068 (95% CI range: 1.03–1.107; *p* < 0.001), while the *TGFB2* by age interaction had an HR of 1.041 (95% CI range: 0.982–1.103; *p* = 0.18), and the age by sex interaction showed an HR of 0.949 (95% CI range: 0.9–1; *p* = 0.048). There was not a significant decrease in HR for the *TGFB2^highMe^* group of patients (HR = 0.067, 95% CI range: 0.002–2.658; *p* = 0.15), *TGFB1^highMe^* (HR = 0.623, 95% CI range: 0.284–1.368; *p* = 0.239), or *TGFB3^highMe^* (HR = 1.041, 95% CI range: 0.54–2.008; *p* = 0.904), whereas a significant decrease in HR was observed for the *MGMT^highMe^* group (HR = 0.452, 95% CI range: 0.243–0.841; *p* = 0.012) (Figure 7).

## 4. Discussion

### 4.1. Identification of Reactomes Enriched with mRNA Expression of Genes Negatively Correlated with TGFB2 Gene Methylation

The striking finding of this study was that *TGFB2* methylation improved survival hazard ratios over a well-established *MGMT* methylation as a prognostic marker in young adult male GBM patients. We sought to understand at the pathway level by examining Reactome pathways correlated with *TGFB2* gene methylation using gene set enrichment analysis (GSEA) and quantifying normalized enrichment scores (NESs) for 1272 Reactome pathways. We identified 64 pathways that showed significant negative correlation across all patient subsets. Eight of these Reactome pathways collectively represented the complex signaling cascade that occurs following TCR engagement, leading to T-cell activation, differentiation, and effector functions (phosphorylation of CD3 and TCR zeta chains12; translocation of ZAP-70 to immunological synapse; generation of second messenger molecules; downstream TCR signaling; CD28-dependent Vav1 pathway; co-stimulation by the CD28 family; PD-1 signaling; and Interleukin-2 family signaling) [16,17].

The examination of the gene-level mRNA upregulation greater than 5-fold in tumor tissues showed the following genes: HLA-DQA1 (37.3-fold increase in tumor tissue, *p* < 0.0001); PDCD1LG (32.9-fold increase, *p* < 0.0001); CD3D (24.1-fold increase, *p* < 0.0001); HLA-DRB1 (18.2-fold increase, *p* < 0.0001); CD86 (16.5-fold increase, *p* < 0.0001); HLA-DPA1, (15.9-fold increase, *p* < 0.0001); HLA-DQB1 (11.8-fold increase, *p* < 0.0001); HLA-DPB1 (10-fold increase, *p* < 0.0001); CD3E (8.6-fold increase, *p* < 0.0001); LYN (6.7-fold increase, *p* < 0.0001); PDCD1 (5.6-fold increase, *p* < 0.0001); PTPRC (5.6-fold increase, *p* < 0.0001); and PTPN6 (5.5-fold increase, *p* < 0.0001).

Recent studies have demonstrated that (T-cell receptor) TCR complex activation contributes to the immune response against gliomas [18]. These studies showed TCR responses during the expansion of glioma-infiltrating lymphocytes (GIL), which help regain cellular fitness upon in vitro expansion [18], to be potentially valuable for augmenting immunotherapies applied to gliomas. Furthermore, paired ex vivo single cell (sc)TCR/RNA-seq and post-expansion TCRB-seq have revealed predictive transcriptional signatures that can determine GIL expansion [18], including Granzyme A (GrzA), Granzyme H (GrzH), Chemokine ligand 5 (CCL5), Natural killer cell granule protein 7 (NKG7), and Granulysin (GNLY), and were highly expressed on expander T-cells. In the context of therapeutic applications, TCR-engineered adoptive cell therapy has shown effectiveness in treating intracranial murine glioblastoma, highlighting the potential of adoptively transferred neoantigen-specific T-cells [19]. Our findings suggest that high levels of *TGFB2* gene methylation or knockdown of TGFB2 mRNA expression are hypothesized to augment TCR-mediated cellular responses to aid immunotherapies using these types of adoptive cell therapies [20,21].

### 4.2. Identification of Reactomes Enriched with mRNA Expression of Genes Negatively Correlated with MGMT Gene Methylation

Since the multivariate Cox proportional hazards model suggested independent effects of *TGFB2* and *MGMT* methylation, we examined Reactomes that showed differential enrichment in the mRNA expression of genes correlated with *MGMT* compared to *TGFB2* methylation (Appendix A) for GBM patients and subsets stratified according to sex (all patients (N = 101), male (N = 59), and female (N = 42)). These 31 pathways were represented by 327 genes, of which 45 predicted a significant negative correlation with *MGMT* methylation (*p* < 0.05). The Anaphase-Promoting Complex/Cyclosome (APC/C) processes featured prominently in the Reactome descriptions for the mRNA expression correlated with *MGMT* methylation (eight Reactomes), suggesting biochemical functional differences in *TGFB2* and *MGMT* methylation in GBM patients. Out of the 45 genes negatively correlated with *MGMT* methylation, 39 predicted significant upregulation in tumor versus normal tissue (*p* < 0.0001) (Appendix A). Application of the multivariate Cox proportional hazards model revealed that the mRNA expression of 33 MGMT methylation-correlated genes predicted a significant impact of MGMT methylation (*p* < 0.01), controlling for *TGFB1/2/3* methylation, age at diagnosis, and sex (Appendix A). Only PSMA4 expression predicted a significant positive prognostic impact (HR = 0.54 (0.307–0.948), *p* = 0.032) (Appendix A).

### 4.3. Genes Correlated with TGFB2 Methylation and with Positive Prognostic Impact on OS for GBM Patients

The multivariate model that controlled for genes for *TGFB* ligands and *MGMT* methylations revealed that the mRNA expression of 19 mRNA products (BCL10, CAPZA1, HIF1A, HLA-DRB1, IFI35, IFIH1, IFIT3, IRAK4, ITCH, NUDT12, PARP9, PSMA4, RSAD2, SNAP23, STAT1, TRIM21, TRIM22, TRIM5, and UBA3) was positively prognostic, with increasing levels of mRNA expression showing decreasing hazard ratios (Appendix A). We validated bulk RNA-seq from the TCGA study using the expression of the genes in an independent dataset utilizing RNA sequencing quantification from single cells obtained from 20 adult and 8 pediatric *IDH*-wildtype GBM patients (Appendix A and Figure 4B,C). Eleven genes were preferentially expressed in the adult samples (PARP9, IFI35, TRIM22, STAT1, TRIM5, HIF1A, RSAD2, IFIT3, IRAK4, ITCH, and MALT1) (Appendix A, Figure 4B,C). Of these genes, IFI35, TRIM22, STAT1, TRIM5, RSAD2, and IFIT3 were interferon related, and HIF1A was Interleukin related. In the adult patients, of the genes predicting positive prognosis at increasing levels of mRNA expression, PARP9 and IFI35 predicted high levels of expression in the macrophages; TRIM22 was highly expressed in the macrophages and T-cells; TRIM5 and HIF1A were highly expressed in the macrophages and malignant cells; IRAK4 was highly expressed in the macrophages, malignant cells, and T-cells (Appendix A and Figure 4B,C). Four genes were highly expressed in the majority of the macrophages surveyed and predicted a positive prognostic impact (HIF1A: 89.26% of macrophage cells; TRIM22: 80.5%; IRAK4: 92.18%; PARP9: 50.8%), suggesting an antitumor role for M1-like macrophages expressing these four markers. A Kaplan–Meier analysis of GBM patients, stratified by methylation load and mRNA expression (Figure 5B–D), demonstrated a statistically significant enhancement in survival durations for the patients predicting elevated levels of HIF1A, TRIM22, and PARP9 mRNA, coupled with a high methylation load (sum of beta values for *TGFB1/2/3* and *MGMT*). This finding indicates a prognostically relevant interaction between gene methylation and the mRNA expression of these genes as expressed in macrophages.

HIF1A (hypoxia-inducible factor 1-alpha) has been associated with therapy resistance and poor prognosis in glioblastoma multiforme. Bou-Gharios et al. (2024) found that overexpression of HIF1A correlates with worse patient outcomes and resistance to treatment [22]. Additionally, Lo Dico et al. (2018) reported that HIF-1α expression decreases in cells that respond to temozolomide (TMZ) but remains unchanged in resistant cells, suggesting that HIF-1α plays a critical role in maintaining resistance to this chemotherapy [23]. This dual role of HIF1A in promoting malignancy and inhibiting effective treatment underscores its importance as a therapeutic target in GBM. Targeting HIF-1α has been the subject of clinical trials for the treatment of gliomas, with limited results [24,25,26].

TRIM22 (tripartite motif containing 22) is significantly overexpressed in GBM patients, and its knockout reduces tumor growth while enhancing sensitivity to temozolomide (TMZ) [27]. It promotes GBM cell proliferation by regulating SPHK2 and modifying MAPK signaling components [27]. Elevated TRIM22 levels correlate with aggressive glioma characteristics and activate NF-κB signaling [28]. It is also linked to increased immune cell infiltration in GBM and low-grade gliomas, with potential inhibitory effects on glioma progression through downregulation along the PI3K/AKT pathway [29]. High TRIM22 expression indicates poor survival outcomes and correlates with various WHO glioma grades and immune cell types [30].

In glioblastomas, IRAK4 (Interleukin 1 Receptor Associated Kinase 4) is upregulated after temozolomide (TMZ) treatment in glioma cells and mediates Toll-like receptor signaling and chemoresistance [31]. The authors also report that IRAK4 knockdown increased TMZ resistance, while high IRAK4 levels induced by TMZ led to IRAK1 downregulation and NFkB pathway inhibition [31]. Wang et al. (2021) reported that IRAK4 was found to be significantly overexpressed in glioma compared to normal brain tissue and that high IRAK4 expression correlated with advanced WHO grade, *IDH* wildtype, and 1p19q non-co-deletions, establishing a link between IRAK4 overexpression and decreased survival rates in glioma patients, suggesting it may be a potential oncogene regulating cell signaling pathways in glioma [32]. IRAK4 inhibitor emavusertib is currently being assessed in phase 1/2 clinical studies for hematologic cancers and several solid tumors [33]. Research has shown that IRAK4 inhibition can have anticancer effects, synergize with current treatments, and potentially enhance antitumor immunity. These findings may encourage future clinical trials exploring IRAK4 inhibition in glioma treatment [33].

PARP9 (poly(ADP-Ribose) polymerase family member 9) has been identified as a potential biomarker and therapeutic target for gliomas. Studies have shown that PARP9 is highly expressed in glioma tissues compared to normal brain tissue, and its high expression is associated with poor prognosis and advanced clinicopathological features [34]. In contrast, our analysis suggested a positive prognostic impact of PARP9 when considering methylation states of *TGFB1/2/3* and *MGMT*, sex, age, and age interactions. The single-cell analysis revealed a high level of PARP9 expression in pro-inflammatory M1-like macrophages that may mitigate the negative prognostic impact observed in previous studies. Iwata et al. (2016) identified PARP9 and PARP14 as key regulators of macrophage activation, with PARP9 promoting pro-inflammatory responses and enhancing STAT1-dependent gene expression, while PARP14 inhibits these processes. Their study demonstrates that PARP9 overexpression fosters a pro-inflammatory state through increased STAT1 signaling via ADP-ribosylation. This interplay between PARP9 and PARP14 is pivotal in defining the inflammatory phenotype of macrophages, highlighting their potential as therapeutic targets for inflammatory diseases [35].

Therefore, the complex interplay between pro-inflammatory factors, mRNA expression in malignant and immune cells, and gene methylation will need to be considered to fully evaluate the four targets (HIF1A, TRIM22, IRAK4, and PARP9) and select patients who would, at best, benefit from combination therapies targeting the immune system.

The expression of TRIM5 (Tripartite Motif Containing 5) mRNA also predicted a prognostic impact (HR = 0.607 (95% CI range: 0.413–0.892; *p* = 0.011)) in which significant impacts were observed for *TGFB2* methylation, where the HR for *TGFB2^highMe^* was 0.016 (95% CI range: 0–0.919; *p* = 0.045), and the age by sex interaction (*p* = 0.046) (Figure 6). TRIM5α protein is important in innate immunity and antiviral defense. TRIM5 recognizes and binds to retroviral particles, disrupting and degrading viral capsid proteins in the cytoplasm [36]. TRIM5α also interacts with the TGF-β-activated kinase 1 (TAK1) complex, leading to the activation of signaling pathways that stimulate innate immunity [37,38]. TRIM5 is upregulated in glioma tissues compared to normal brain tissues [30,39], and higher TRIM5 expression is associated with poor prognosis [30,39]. Interestingly, CD2AP (CD2-associated protein) interacts with TRIM5 to activate NF-κB signaling, promoting GBM malignancy [40]. In addition, TRIM proteins (TRIM11/25/26/28/33/59/62/66/72) and TGFB pathways (Smad4 degradation) have been shown to promote epithelial–mesenchymal transition, cell migration, and invasion of cancer cells through activation of the TGF-β signaling pathway [41]. In contrast, our study showed that the correlation of TRIM5 mRNA with TGFB2 gene methylation reveals the positive prognostic impact of TRIM5 mRNA.

### 4.4. MALT1 mRNA Expression Showed a Negative Prognostic Impact on OS in GBM Patients

MALT1 predicted a 2.1-fold increase in mRNA expression in tumors compared to normal tissues. This gene was the only non-interferon-related gene out of the 20 that showed prognostic impact in GBM patients. The examination of the single-cell RNA-seq data (Figure 4 and Appendix A) showed that the expression in macrophages (30.5%) and T-cells (25.5%) was less than 50% of the total cells surveyed. The multivariate model that included MALT1 mRNA showed that there was a significant increase in HR for MALT1 mRNA expression (HR (95% CI range) = 1.997 (95% CI range: 1.1–3.625; *p* = 0.023)), representing a 2-fold increase in HR for a 2-fold increase in MALT1 mRNA expression. The *TGFB2* methylation by age interaction did not achieve statistical significance (HR (95% CI range) = 1.041 (95% CI range: 0.982–1.103; *p* = 0.18)), while the age by sex interaction effect was significant (HR (95% CI range) = 0.949 (95% CI range: 0.9–1; *p* = 0.048)) (Figure 7). MALT1 is a key signaling molecule in both innate and adaptive immunity [42,43]. It activates T-cells, B-cells, myeloid cells, and NK cells by regulating the NF-κB signaling pathway, driving immune cell activation, proliferation, survival, and cytokine production [42,43,44]. MALT1 is part of the CARMA1-BCL10-MALT1 (CBM) signalosome complex that facilitates antigen receptor signaling in lymphocytes and acts as a paracaspase, cleaving specific substrates to sustain NF-κB signaling, enhancing T-cell adhesion, and stabilizing mRNA for pro-inflammatory genes [42,44,45]. It is critical for developing regulatory T-cells, maintaining immune tolerance, and mediating innate immune responses via receptors such as dectin-1 [42,43]. Dysregulated MALT1 activity is linked to lymphoid malignancies (e.g., MALT lymphoma), autoimmune diseases, and immunodeficiencies [42,44,45]. MALT1 plays an important role in cancer, including glioblastoma, by driving tumor progression through both cell-intrinsic and cell-extrinsic mechanisms [45,46]. It promotes tumor cell survival, proliferation, and invasion via EGFR-induced NF-κB activation, while its scaffolding and protease activities regulate key signaling pathways, including the inactivation of NF-κB pathway inhibitors, specifically A20 and CYLD [45,46]. MALT1 also influences the tumor microenvironment by maintaining immune-suppressive regulatory T-cells. In GBM, MALT1 inhibition, such as with MI-2, leads to reduced tumor cell proliferation, migration, and invasion, induces G1 cell cycle arrest, and enhances immune reactivity in the tumor microenvironment, making it a promising therapeutic target [47]. Recent studies indicate that MALT1 is a downstream target of the canonical TGF-β/Smad3 pathway, where TGF-β stimulation increases MALT1 expression dose- and time-dependently via Smad3 phosphorylation. This relationship is supported by correlated expressions of SMAD3 and MALT1 in cancer data (TCGA). MALT1 also mediates TGF-β-induced NF-κB signaling, facilitating nuclear translocation of NF-κB p65 and transcriptional activation, with the CBM signalosome (CARMA-BCL10-MALT1) playing a role in activating NF-κB target genes. This MALT1-TGF-β-NF-κB axis explains TGF-β’s dual role in cancer progression, contributing to survival, proliferation, and metastasis of cancer cells, and highlights MALT1 as a potential therapeutic target for disrupting this interaction in cancer treatment [48].

The existing limitations of these studies stem from the reliance on bulk mRNA expression levels and methylation data for identifying potential prognostic markers in glioblastoma (GBM) patients. This approach may not strongly correlate with protein levels and, therefore, may impact specific biological functions. Future studies will require confirmation of mRNA levels validated using qPCR methods, protein levels, and signaling mechanisms to expand on these initial findings, especially utilizing *TGFB2* and *MGMT* methylation screens from patient-derived tumor tissues.

## 5. Conclusions

We implemented a multivariate Cox proportional hazards model to directly compare hazard ratios for *TGFB1/2/3* and *MGMT* methylation in relation to OS. We considered male versus female, age at diagnosis, and age interactions with TGFB2 gene methylation and sex variables. GBM patients with high levels of *TGFB2* gene methylation predicted a significantly improved OS prognosis relative to *TGFB1* methylation and *MGMT* methylation groups of young, male GBM patients. To identify prognostic gene markers, we screened the mRNA expression of 14,364 genes curated across 1272 Reactome pathways to identify consistently affected pathways, comparing young, old, and all GBM patients enriched with the expression of genes negatively correlated with *TGFB2* methylation. We identified the Reactome pathways that collectively represented the complex signaling cascade that occurs following TCR engagement, leading to T-cell activation, differentiation, and effector functions. *TGFB2* gene methylation was negatively correlated with interferon-related pathways and the expression of mRNA for genes involved in the key mechanisms for priming naïve CD8+ T-cells via the presentation of tumor antigens by professional antigen-presenting cells. The expression of four genes was negatively correlated with TGFB2 gene methylation and was highly expressed in macrophages, as assessed using single-cell analysis (HIF1A, TRIM22, IRAK4, and PARP9). These markers were also upregulated in tumor tissues, and the mRNA expression of these genes predicted a significant positive prognostic impact on OS (using the multivariate Cox proportional hazards model accounting for gene methylations, sex, age, and age interactions as variables). Using Kaplan–Meier analysis, the mRNA expression values of three of these genes (HIF1A, TRIM22, and PARP9 mRNA) showed prolonged improvements in survival times for patients expressing high levels of these three genes along with high methylation load across all the GBM patients. TRIM5 mRNA also predicted a prognostic impact in which significant impacts were observed for *TGFB2* methylation, *MGMT* methylation, and the age by sex interaction, suggesting that high levels of TRIM5 mRNA, *MGMT* methylation, and *TGFB2* methylation levels can be used as positive prognostic indicators for GBM patients. MALT1 mRNA expression was the only gene product with a negative prognostic impact on OS in the GBM patients (HR (95% CI range) = 1.997 (1.1–3.625); *p* = 0.023) and, therefore, can be targeted for immune therapies. *TGFB2* gene methylation predicts improved OS in adult GBM patients that is age and sex-dependent, and correlation with mRNA identifies Reactome pathways that could potentially remodel the tumor microenvironment to augment T-cell activation and antigen processing, which would be potentially beneficial to patient survival for those undergoing immune therapies. We have identified a panel of gene methylations and mRNA expression markers that predict prolonged survival for GBM patients.

## Figures and Tables

**Figure 1 cancers-17-01122-f001:**
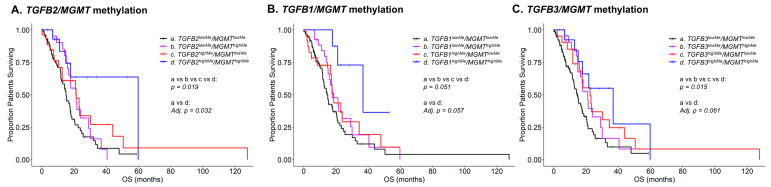
Subsets of young GBM patients with high *TGFB1/2/3* and *MGMT* methylation levels demonstrate improved OS. GBM patients were interrogated using the TCGA dataset, and OS was correlated with methylation beta values (upper quartile percentile cut-off for high methylation levels; superscripted “highMe” compared to low methylation; “lowMe” for each of the genes) for young patients (median cut-off for young patients < 60 years for both male and female patients) to investigate the prognostic impact of patients with combinations of high and low levels of *TGFB1/2/3* and *MGMT* methylations. Kaplan–Meier analysis was performed for four combinations of *TGFB1/2/3* and *MGMT* methylation. To assess significance, the survival times across the 4 curves were determined, and the pairwise test between patients with high levels of both *TGFB1/2/3* and *MGMT* versus low levels of both *TGFB1/2/3* and *MGMT* methylations (Log-rank Adjusted *p*-values (Adj. *p*) using the Benjamini–Hochberg correction for six multiple comparisons across the four groups of patients. (**A**) The median survival time for 15 patients with high levels of *TGFB2* and *MGMT* methylation (*TGFB2^highMe^*/*MGMT^highMe^*: mOS = 60 (95% CI: 17.8–NA, death events = 5) months) was significantly longer than for patients with low levels of *TGFB2* and *MGMT* methylation (*TGFB2^lowMe^*/*MGMT^lowMe^*: N = 75; mOS = 15 (95% CI: 12.2–18.1, death events = 50) months; Adj. *p* = 0.032). There was a significant trend of increasing mOS across all four subsets of GBM patients (*p* = 0.019). Patients (N = 22) with *TGFB2^lowMe^*/*MGMT^highMe^* predicted mOS of 21.1 (95% CI: 16.6–36.9, death events = 17) months, and patients (N = 26) with *TGFB2^highMe^*/*MGMT^lowMe^* predicted mOS of 21.2 (95% CI: 12.6–NA, death events = 17) months. (**B**) Patients with *TGFB1* and *MGMT* methylation showed an increasing trend of mOS related to increasing methylation levels (*p* = 0.051): The median survival time for *TGFB1^lowMe^*/*MGMT^lowMe^* group of 75 patients was 14.5 (95% CI: 12.2–20.3, death events = 53) months; mOS for *TGFB1^lowMe^*/*MGMT^highMe^* group of 29 patients was 17.9 (95% CI: 15.1–30, death events = 19) months; mOS for *TGFB1^highMe^*/*MGMT^lowMe^* group of 26 patients was 18.6 (95% CI: 16.7–NA, death events = 14) months; and mOS for *TGFB1^highMe^*/*MGMT^highMe^* group of 8 patients was 36.9 (95% CI: 21.1–NA, death events = 3) months. The increase in the mOS difference between patients with high levels of *TGFB1* and *MGMT* methylation compared to those with low levels of *TGFB1* and *MGMT* methylation was borderline significant (14.5 versus 36.9 months, respectively; Adj. *p* = 0.051). (**C**) Patients with *TGFB3* and *MGMT* methylation showed a significant increasing trend of mOS related to increasing methylation levels (*p* = 0.015): The mOS for *TGFB3^lowMe^*/*MGMT^lowMe^* group of 80 patients was 14.5 (95% CI: 12.2–18.6, death events = 51) months; the mOS for *TGFB3^lowMe^*/*MGMT^highMe^* group of 23 patients was 21.1 (95% CI: 16.6–NA, death events = 15) months; the mOS for *TGFB3^highMe^*/*MGMT^lowMe^* group of 21 patients was 22.5 (95% CI: 15–NA, death events = 16) months; and the mOS for *TGFB3^highMe^*/*MGMT^highMe^* group of 14 patients was 36.9 (95% CI: 17.8–NA, death events = 7) months. The increase in the mOS difference between patients with high levels of *TGFB3* and *MGMT* methylation compared to those with low levels of *TGFB3* and *MGMT* methylation was borderline significant (14.5 versus 36.9 months, respectively; Adj. *p* = 0.061).

**Figure 2 cancers-17-01122-f002:**
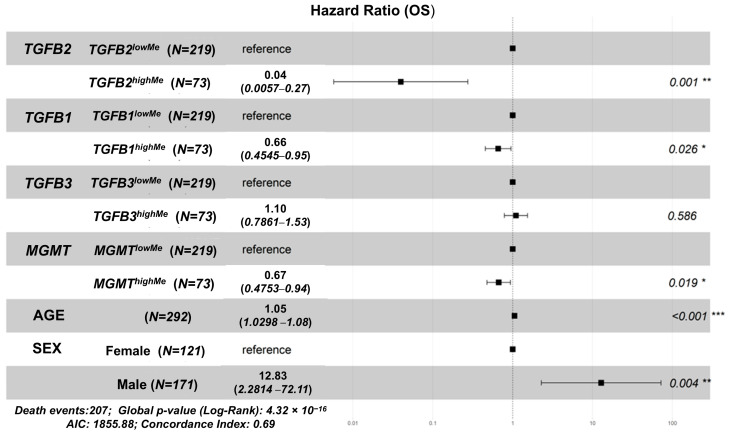
High levels of *TGFB1/2/3 and MGMT* methylation demonstrate a positive prognostic impact on OS in GBM patients. A Cox proportional hazards model was implemented to assess the individual effects of *TGFB1/2/3* and *MGMT* methylation (upper quartile cut-off) on OS hazard ratios (HR) (N = 292, 207 death events), controlling for age at diagnosis (AGE), sex (SEX; male relative to female), and interaction terms for *TGFB2* × age and age × sex interactions. The regression model controlled for the significant impacts of age at diagnosis (AGE; HR (95% CI range) = 1.053 (1.03–1.077); *p* < 0.001) and sex (HR (95% CI range) = 12.826 (2.281–72.108); *p* = 0.004) and controlled for the significant interaction terms for *TGFB2* methylation × age (HR (95% CI range) = 1.052 (1.02–1.085); *p* = 0.001) and age × sex (HR (95% CI range) = 0.966 (0.94–0.993); *p* = 0.014). *** *p* < 0.001, ** *p* < 0.01, * *p* < 0.05.

**Figure 3 cancers-17-01122-f003:**
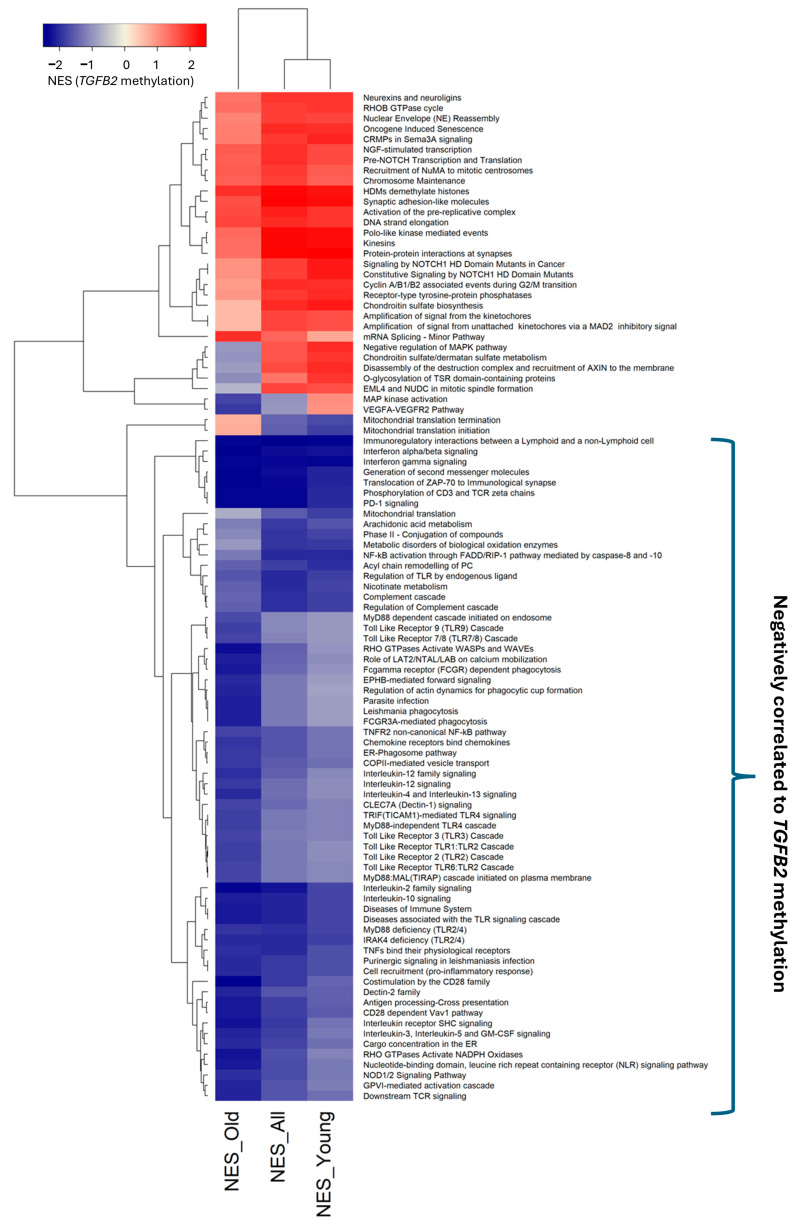
Identification of Reactome pathways negatively correlated with *TGFB2* methylation. Beta-values for *TGFB2* methylation were correlated with mRNA expression levels of 14364 genes for subsets of GBM patients stratified according to age at diagnosis (all patients (N = 101), young (N = 70), old (N = 31); cut-off 69 years). Normalized enrichment scores (NESs) were calculated for 1272 Reactome pathways. Negative NES scores suggested an over-representation of mRNA expression of genes negatively correlated with *TGFB2* gene methylation (each Reactome pathway was populated with between 10 and 100 genes). The cluster figure depicts significantly affected pathways across all subsets of patients (NES scores *p* < 0.001 determined from 100,000 permutations for all or young or old patients). Pathways negatively correlated with *TGFB2* methylation were represented by 1161 genes from 64 pathways, of which 275 genes predicted significant negative correlations with *TGFB2* methylation (*p* < 0.05, FDR = 0.11).

**Figure 4 cancers-17-01122-f004:**
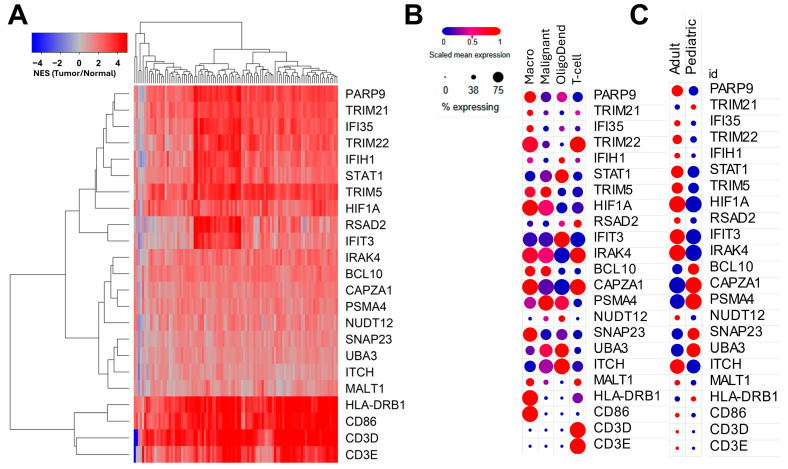
Prognostic genes upregulated in tumors when correlated with *TGFB2* methylation from the TCGA dataset and mRNA expression validation in single-cell determinations in GBM samples. Two hundred and seventy-five genes negatively correlated with *TGFB2* methylation; one hundred and sixty-nine were significantly upregulated in tumor samples (*p* < 0.0001, N = 99 cross-referenced with patients for whom OS data and mRNA expression levels were reported) and were screened for prognostic impact using a multivariate Cox proportion hazards model which measured the effects of mRNA expression of each of the genes (Gene 2), *TGFB1/2/3*, and *MGMT* methylation (upper quartile cut-off) on OS (N = 101, 73 death events), controlling for age at diagnosis (AGE), sex (male relative to female), and interaction terms for *TGFB2* × age and × sex interaction. Twenty genes predicted significant impact for Gene 2 HR (*p* < 0.05, of which nineteen genes predicted positive prognostic impact (FDR = 0.22), and only MALT1 predicted negative prognostic impact). (**A**) The cluster figure depicts log2 transformed patient-level fold-change values for tumor (N = 99 evaluable mRNA levels), mean-centered to expression in normal tissue (N = 1141 from all brain regions) comparisons organized using a two-way hierarchical clustering algorithm (blue to red for increasing fold-change). The figure includes 20 genes predicting prognostic impact and 3 marker genes (CD3D and CD3E for T-cells and CD86 for M1-like macrophages). All 23 genes showed significant upregulation in tumor tissues. We analyzed the expression of the 23 genes in an independent dataset utilizing RNA sequencing quantification from single cells obtained from 20 adult and 8 pediatric *IDH*-wildtype glioblastoma patients by cell-type ((**B**) Macro: macrophage (754 cells), Malignant (6863 cells), OligoDend: oligodendrocyte (219 cells), and T-cells (94 cells)) and GBM type (Adult 5742 cells and pediatric 2188 cells). (**B**) The dot plot depicts the scaled mean expression (blue to red circles) and the percentage of cells expressing the gene for each cell type (circle size) for the 23 genes organized according to the cluster order from the bulk TCGA RNA-seq data. (**C**) The dot plot depicts the relative expression of each of the 23 genes in adult versus pediatric patients. In adult patients, PARP9 and IFI35 predicted high levels of expression in macrophages; TRIM22 and MALT1 were highly expressed in macrophages and T-cells; TRIM5 and HIF1A were highly expressed in macrophages and malignant cells; IRAK4 was highly expressed in macrophages, malignant cells, and T-cells.

**Figure 5 cancers-17-01122-f005:**
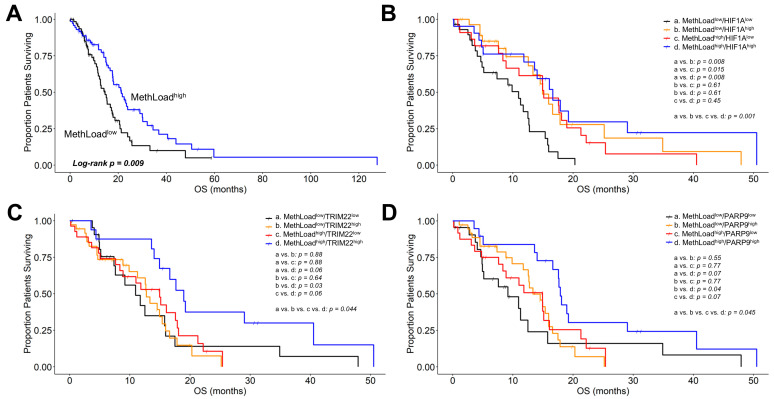
Methylation load and mRNA expression predict favorable prognosis in GBM patients using Kaplan–Meier analysis. OS times were correlated with the combination of methylation beta values (**A**). Median cut-off for high methylation load (MethLoad) calculated using the sum of beta values for *TGFB1/2/3* and *MGMT* genes; superscripted “high” compared to low methylation; “low” and mRNA expression levels for HIF1A (**B**), TRIM22 (**C**), and PARP9 (**D**) using median cut-off values to assign high and low expressing subsets of GBM patients. A combination of methylation load (high and low) and mRNA expression levels (high and low) generated four survival curves for each gene, and the outcomes were compared using log-rank tests, for which the differences in the four curves (a vs. b vs. c vs. d) were determined. The p-values for the pairwise differences were corrected for multiple comparisons using the Benjamini–Hochberg method. (**A**) The mOS time for 72 patients with high methylation load was significantly longer (mOS = 21.3 (95% CI: 17.7–30, death events = 46) months) than the 66 patients with low methylation load (mOS = 14.5 (95% CI: 11.7–17.9, death events = 43) months; Log-rank Chi-square = 6.87, *p*-value = 0.009). (**B**) Examination of the combination of HIF1A mRNA expression levels and methylation load showed the shortest mOS time for the 29 patients with MethLoad^low^/HIF1A^low^ (mOS = 10.9 (95% CI: 5.2–12.8, death events = 24) months) compared to the 29 patients from the MethLoad^low^/HIF1A^high^ group (mOS = 14.9 (95% CI: 13.3–NA, death events = 16) months; Adj. *p* = 0.008), 22 patients with MethLoad^high^/HIF1A^low^ (mOS = 15.1 (95% CI: 8.9–21.3, death events = 19) months; Adj. *p* = 0.015), and 21 MethLoad^high^/HIF1A^high^ patients (mOS = 16.6 (95% CI: 13.6–NA, death events = 14) months; Adj. *p* = 0.008). (**C**) There was a significant trend showing increasing TRIM22 mRNA and methylation load results in improved survival outcomes across the 4 groups of patients (*p* = 0.044): MethLoad^low^/TRIM22^low^ (N = 22; mOS = 10.9 (95% CI: 7.6–17.5, death events = 16) months); MethLoad^low^/TRIM22^high^ (N = 36; mOS = 12.8 (95% CI: 11.3–16.6, death events = 24) months); MethLoad^high^/TRIM22^low^ (N = 27; mOS = 15 (95% CI: 8.4–18.1, death events = 21) months); and MethLoad^high^/TRIM22^high^ (N = 16; mOS = 19 (95% CI: 14.9–NA, death events = 12) months) subsets. A statistically significant difference was observed when MethLoad^low^/TRIM22^high^ was compared with MethLoad^high^/TRIM22^high^ (Adj. *p* = 0.032). (**D**) There was also a significant trend showing increasing PARP9 mRNA and methylation load results in improved survival outcomes across the 4 groups of patients (*p* = 0.045): MethLoad^low^/PARP9^low^ (N = 22; mOS = 9.2 (95% CI: 4.9–NA, death events = 16) months); MethLoad^low^/PARP9^high^ (N = 36; mOS = 13.3 (95% CI: 12.6–16.6, death events = 24) months); MethLoad^high^/PARP9^low^ (N = 24; mOS = 14.9 (95% CI: 8.4–22.2, death events = 18) months); and MethLoad^high^/PARP9^high^ (N = 19; mOS = 17.9 (95% CI: 16.6–NA, death events = 15) months) subsets. A statistically significant difference was observed when MethLoad^low^/PARP9^high^ was compared with MethLoad^high^/PARP9^high^ (Adj. *p* = 0.043).

**Figure 6 cancers-17-01122-f006:**
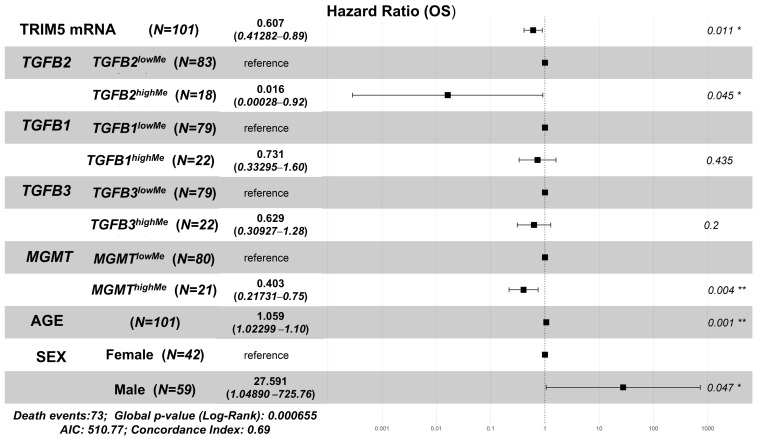
Forest plot depicting the positive prognostic impact of TRIM5 mRNA. The Cox proportional hazards model was implemented to assess the individual effects of TRIM5 mRNA, *TGFB1/2/3,* and *MGMT* methylation (upper quartile cut-off, “highMe”) on OS (N = 101, 73 death events), controlling for age at diagnosis (AGE), sex (male relative to female), and interaction terms for *TGFB2* × age and age × sex interaction. The forest plot depicts the impact on hazard ratio (HR) for each factor. ** *p* < 0.01, * *p* < 0.05.

**Figure 7 cancers-17-01122-f007:**
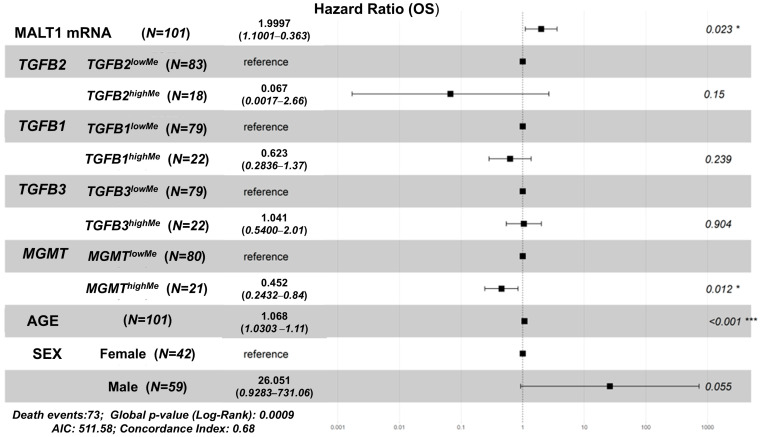
Negative prognostic impact of MALT1 mRNA expression on OS hazard ratios (HR) in GBM patients. The multivariate Cox proportional hazards model was implemented to assess the individual effects of MALT1 mRNA expression (log2 TPM values), *TGFB1/2/3,* and *MGMT*, controlling for age at diagnosis (AGE), sex (male relative to female), and interaction terms for TGFB2 × age and age × sex interactions. The forest plot depicts the impact on hazard ratio (HR) for each factor. *** *p* < 0.001, * *p* < 0.05.

## Data Availability

We analyzed clinical metadata and methylation beta values obtained from normalized values merged between hm27 and hm450 platforms for 297 patients who were diagnosed with GBM (https://www.cbioportal.org/study/summary?id=gbm_tcga_pan_can_atlas_2018; data file: “data_methylation_hm27_hm450_merged.txt” accessed on 22 February 2024). Gene expression values were reported as log2 transformed transcripts per million (TPM) and downloaded as RNAseq data files (https://toil-xena-hub.s3.us-east-1.amazonaws.com/download/TcgaTargetGtex_rsem_gene_tpm.gz; full metadata, accessed on 25 July 2023) from the UCSC Xena web platform (https://xenabrowser.net/datapages/ accessed on 25 July 2023). Single-cell data for GBM patients were obtained from https://singlecell.broadinstitute.org/single_cell/study/SCP393/single-cell-rna-seq-of-adult-and-pediatric-glioblastoma, accessed 27 February 2025.

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
