# Peer review of "Positive Prognostic Overall Survival Impacts of Methylated TGFB2 and MGMT in Adult Glioblastoma Patients"

_cancers, 2025, doi:10.3390/cancers17071122_

Round 1
Reviewer 1 Report
Comments and Suggestions for Authors
In this manuscript, the authors present a novel finding that TGFB2 gene methylation serves as a stronger prognostic marker than MGMT methylation in adult GBM patients, demonstrating a 19-fold improvement in survival risk. It is evident that the authors have invested considerable effort into this work; however, several major points need to be addressed prior to publication:
- In lines 256-258, it states: “The median OS time for 75 patients with high levels of MGMT methylation (MGMThighMe) was 16.8 months (95% CI: 15.3 - 23.9, Events = 46), which was significantly longer than the median OS time for the remaining patients (N=222), at 12.8 months (95% CI: 11.7 - 14.9, Events = 163).” Could you please clarify what is meant by "events"?
- In Figure 7, there appears to be an inconsistency in the background of the second and fourth rows of the first column. Please verify and correct this issue.
- The assertion that TGFB2 and MGMT gene methylation are involved in T-cell activation and antigen presentation requires further substantiation; therefore, I recommend analyzing single-cell data to provide additional evidence supporting this claim.
Author Response
In this manuscript, the authors present a novel finding that TGFB2 gene methylation serves as a stronger prognostic marker than MGMT methylation in adult GBM patients, demonstrating a 19-fold improvement in survival risk. It is evident that the authors have invested considerable effort into this work; however, several major points need to be addressed prior to publication:
We thank the reviewer for acknowledging our effort on this manuscript. We feel the suggestions enhanced the publication.
- In lines 256-258, it states: “The median OS time for 75 patients with high levels of MGMT methylation (MGMThighMe) was 16.8 months (95% CI: 15.3 - 23.9, Events = 46), which was significantly longer than the median OS time for the remaining patients (N=222), at 12.8 months (95% CI: 11.7 - 14.9, Events = 163).” Could you please clarify what is meant by "events"?
Response 1: We have clarified the meaning of events to indicate “death events”
- In Figure 7, there appears to be an inconsistency in the background of the second and fourth rows of the first column. Please verify and correct this issue.
Response 2: We have fixed the formatting issues regarding the background to these figures
- The assertion that TGFB2 and MGMT gene methylation are involved in T-cell activation and antigen presentation requires further substantiation; therefore, I recommend analyzing single-cell data to provide additional evidence supporting this claim.
Response 3: We thank the reviewer for this recommendation, and we have clarified the prominent role of the M1-like macrophage markers in GBM by validating the prognostic markers discovered from the TCGA bulk mRNA markers. Two figures were added to the manuscript in addition a supplementary figure:
“Figure 5. Prognostic genes upregulated in tumors when correlated with TGFB2 methylation from the TCGA dataset and mRNA expression validation in single-cell determinations in GBM samples.”
“Figure 6. Prognostic impact of genes highly expressed in adult macrophages identified from bulk and single-cell RNA-seq experiments.”
“Figure S2. Clustering using a tSNE projection of cells isolated from wild-type GBM patients.”
The updated figures led to an update of the discussion of the prognostic markers.
“The multivariate model that controlled for genes for TGFB ligands and MGMT methylations revealed that the mRNA expression of 19 mRNA products (BCL10, CAPZA1, HIF1A, HLA-DRB1, IFI35, IFIH1, IFIT3, IRAK4, ITCH, NUDT12, PARP9, PSMA4, RSAD2, SNAP23, STAT1, TRIM21, TRIM22, TRIM5, UBA3) was positively prognostic, with increasing levels of mRNA expression showing decreasing hazard ratios (Table S3). We validated bulk RNA-seq from the TCGA study using the expression of the genes in an independent dataset utilizing RNA sequencing quantification from single cells obtained from 20 adult and 8 pediatric IDH-wildtype Glioblastoma patients (Figure S2, Figure 5B and 5C). Eleven genes were preferentially expressed in adult samples (PARP9, IFI35, TRIM22, STAT1, TRIM5, HIF1A, RSAD2, IFIT3, IRAK4, ITCH, MALT1) (Figure S2, Figure 5B and 5C). Of these genes, IFI35, TRIM22, STAT1, TRIM5, RSAD2, and IFIT3 were interferon related, and HIF1A was Interleukin related. In Adult patients, of the genes exhibiting positive prognosis at increasing levels of mRNA expression PARP9 and IFI35 exhibited high levels of expression in Macrophages; TRIM22 was highly expressed in Macrophages and T-cells; TRIM5 and HIF1A were highly expressed in Macrophages and Malignant cells; IRAK4 was highly expressed in Macrophages, Malignant and T-cells (Figure S2, Figures 5B and 5C).
Four genes were highly expressed in the majority of macrophages surveyed that exhibited positive prognostic impact (HIF1A: 89.26% of macrophage cells; TRIM22: 80.5%; IRAK4 92.18%; PARP9: 50.8%) (Figure 6) suggesting an antitumor role for M1-like macrophages expressing these four markers.” (Section 4.1).
We have modified the title to reflect the role of macrophage markers as prognostic indicators.
“Positive Prognostic OS impact of TGFB2 and MGMT Gene Methylation correlated with mRNA for Macrophage markers in Adult GBM Patients.”
Reviewer 2 Report
Comments and Suggestions for Authors
The authors analyzed publicly available datasets from 297 glioblastoma patients, correlating clinical data with gene methylation profiles to identify independent prognostic biomarkers. Their findings confirm that MGMT methylation correlates with overall survival (OS) after standard therapy. Notably, TGFB2 methylation was associated with even greater benefit. Additionally, several immune-related genes and pathways were linked to improved survival, suggesting potential targets for immunotherapy.
The 2021 WHO classification of nervous system tumors introduced a distinction between astrocytoma WHO grade IV and glioblastoma, redefining the latter as IDH-wildtype. Since the dataset analyzed (from 2018) predates this classification, the authors should discuss whether their findings, particularly correlations with longer OS, may be influenced by the inclusion of IDH-mutant astrocytomas.
Specific comments:
- L. 2: Consider shortening the title for better readability.
- L. 24: Verify and correct (or rephrase) the reported percentage, which appears too high.
- L. 826: Check the accuracy of the provided link (spelling?) or clarify how to access the dataset:
https://www.cbioportal.org/study/summary?id=gbm_tcga_pan_can_atlas_2018.
Since IDH-mutant astrocytoma IV has a better prognosis than IDH-wildtype glioblastoma IV, the authors should ensure that IDH-mutant cases are excluded when referring to glioblastoma. The observed findings might be influenced by IDH status and should be discussed accordingly.
It may also be interesting to compare these findings on glioblastomas with other tumor types not typically associated with low MGMT levels. For instance, a subset of breast cancers exhibits similarly low MGMT expression (DOI: 10.1002/ijc.2910610308).
Author Response
Specific comments:
- L. 2: Consider shortening the title for better readability.
Response 1: We have shortened the title to “Positive Prognostic OS impact of TGFB2 and MGMT Gene Methylation correlated with mRNA for Macrophage markers in Adult GBM Patients.”
- 24:Verify and correct (or rephrase) the reported percentage, which appears too high.
Response 2: We thank the reviewer for this observation and corrected the manuscript accordingly, “common primary malignant brain tumor in adults, constituting 45.6% of tumors.”
- L. 826: Check the accuracy of the provided link (spelling?) or clarify how to access the dataset:
https://www.cbioportal.org/study/summary?id=gbm_tcga_pan_can_atlas_2018.
Response 3: The link provided does navigate to the appropriate web page. We tried it from within the Word document or by cutting and pasting it to the web browser. We have added another link to the manuscript to provide an additional method.
(https://www.cbioportal.org/study/summary?id= gbm_tcga_pan_can_atlas_2018; data file: “data_methylation_hm27_hm450_merged.txt” accessed on 22 February 2024, or can be accessed via https://www.cbioportal.org/datasets and then click onto Glioblastoma Multiforme (TCGA, PanCancer Atlas).
Since IDH-mutant astrocytoma IV has a better prognosis than IDH-wildtype glioblastoma IV, the authors should ensure that IDH-mutant cases are excluded when referring to glioblastoma. The observed findings might be influenced by IDH status and should be discussed accordingly.
Response 4: We reanalyzed the dataset and produced a new set of figures in the revised manuscript:
Figure 1. We removed 5 IDH mutant patients from the analysis to evaluate 292 patients (originally 297 patients). The main conclusions were unaffected. We reported the updated statistics for the high versus low methylation comparisons:
“GBM patients (N=73) with high levels of TGFB2 methylation (TGFB2highMe) exhibited a median OS time of 14 (95% CI: 12.4 - 23.1, death events = 50) months, which was similar to the median OS time of 14.4 (95% CI: 12.5 - 15.3, death events = 157) months for 219 patients with lower methylation levels (TGFB2lowMe). At later time points (> 20 months), the high and low methylation groups significantly diverged in this Kaplan-Meier analysis with a better prognosis for the TGFB2highMe group of patients (Log-rank Chi-Square = 3.971, P = 0.046) (Figure 1A). The median OS time for 73 patients with high levels of MGMT methylation (MGMThighMe) was 16.8 (95% CI: 15.1 - 21.7, death events = 45) months and was significantly longer than the median OS time for the remaining patients (N=219, 12.7 (95% CI: 11.5 - 14.9, death events = 162) months; Log-rank Chi-Square = 7.729, P = 0.005) (Figure 1B). GBM patients (N=73) in the TGFB1highMe group exhibited an OS time of 17.2 (95% CI: 12.2 - 23.1, death events = 42) months, which was longer than the remaining patients but did not reach statistical significance (N=219; 13.6 (95% CI: 12.5 - 15, death events = 165) months; Log-rank Chi-Square = 3.804, P = 0.051) (Figure 1C). Similar outcomes were observed for the TGFB3highMe group of GBM patients, whereby the median OS time for the high methylation group was 15.0 (N=73; 95% CI: 11.8 - 21.7, death vents = 53) months compared to 14.0 (95% CI: 12.5 - 15.3, Events = 154) months for the 219 remaining patients (Log-rank Chi-Square = 2.22, P-value = 0.136) (Figure 1D).”
Figure 2. The calculated hazard ratios were marginally affected, whereby the improved prognosis of TGFB2 methylation reduced from 19-fold to 17-fold in the updated analysis
“When controlled for Age and Sex, GBM patients with high levels of TGFB2 gene methylation (TGFB2highMe) showed a significantly improved OS prognosis (HR (95% CI range) = 0.04 (0.006-0.274); P = 0.001) - a 17-fold improvement relative to TGFB1highMe (HR (95% CI range) = 0.657 (0.454-0.951); P = 0.026) and MGMThighMe (HR (95% CI range) = 0.667 (0.475-0.936); P = 0.019) groups of GBM patients. Methylation of TGFB3 (TGFB3highMe) did not have a significant impact on OS (HR (95% CI range) = 1.097 (0.786-1.531); P = 0.586) (Figure 2). “
Figure 3. The predicted OS curves were very similar to those of the updated GBM dataset compared with the previous analysis.
“This multivariate impact of Age, Age interaction with TGFB2 methylation, and Sex on OS showed that when separated out by age, young females and young males with high levels of TGFB2 methylation exhibited improved OS outcomes (Young males and young females with high levels of TGFB2 methylation exhibited median OS time > 125 months) Figure 3A and 3B), thereby reducing the sexual dimorphism dependent on TGFB2 methylation compared to TGFB1 (Figure 3C and 3D) and MGMT (Figure 3E and 3F) methylation which exhibited strong sexual dimorphism as shown by large difference in the OS curves for young males and young females (Young males with high levels of MGMT methylation exhibited median OS time of 19.5 months (Figure 3E), this contrasted with Young Females with high levels of MGMT methylation exhibiting a median OS of 127.6 months (Figure 3F)). This divergent response suggested that young males benefit from the high methylation of TGFB2 but not from the high methylation of MGMT (Figure 3).”
Figure 4. The original dataset of 106 patients for whom clinical, methylation and mRNA expression was provided was reduced to 101 for the Reactome analysis. The updated analysis identified additional pathways featuring Toll-like receptors, but the correlation of CXCL10 to TGFB2 methylation was insignificant, and all further analyses of CXCL10 were removed from the updated gene set.
“Beta-values for TGFB2 methylation were correlated with mRNA expression levels of 14364 genes for subsets of GBM patients stratified according to age at diagnosis (All patients (N=101), Young (N=70) Old (N=31; Cut-off 69 yr). Normalized Enrichment Scores (NES) were calculated for 1272 Reactome Pathways. Negative NES scores suggested an over-representation of mRNA expression of genes negatively correlated with TGFB2 gene methylation (each Reactome pathway was populated between 10-100 genes). The cluster figure depicts significantly affected pathways across all subsets of patients (NES scores P<0.001 determined from 100000 permutations for All or Young or Old patients). Pathways negatively correlated to TGFB2 methylation were represented by 1161 genes from 64 pathways, of which 275 genes exhibited significant negative correlations to TGFB2 methylation (P<0.05, FDR = 0.11).”
Figure 5. We identified mRNA expression of 20 genes exhibited a prognostic impact on OS on the updated GBM patients. In addition, we validated the expression of these genes using an independent dataset reporting single-cell RNA seq results from wild-type IDH GBM patients that highlighted the prominent role of macrophages and interferon-related genes contributing to the prognostic impact observed in the TCGA dataset.
“The multivariate model that controlled for genes for TGFB ligands and MGMT methylations revealed that the mRNA expression of 19 mRNA products (BCL10, CAPZA1, HIF1A, HLA-DRB1, IFI35, IFIH1, IFIT3, IRAK4, ITCH, NUDT12, PARP9, PSMA4, RSAD2, SNAP23, STAT1, TRIM21, TRIM22, TRIM5, UBA3) was positively prognostic, with increasing levels of mRNA expression showing decreasing hazard ratios (Table S3). We validated bulk RNA-seq from the TCGA study using the expression of the genes in an independent dataset utilizing RNA sequencing quantification from single cells obtained from 20 adult and 8 pediatric IDH-wildtype Glioblastoma patients (Figure S2, Figure 5B and 5C). Eleven genes were preferentially expressed in adult samples (PARP9, IFI35, TRIM22, STAT1, TRIM5, HIF1A, RSAD2, IFIT3, IRAK4, ITCH, MALT1) (Figure S2, Figure 5B and 5C). Of these genes, IFI35, TRIM22, STAT1, TRIM5, RSAD2, and IFIT3 were interferon related, and HIF1A was Interleukin related. In Adult patients, of the genes exhibiting positive prognosis at increasing levels of mRNA expression PARP9 and IFI35 exhibited high levels of expression in Macrophages; TRIM22 was highly expressed in Macrophages and T-cells; TRIM5 and HIF1A were highly expressed in Macrophages and Malignant cells; IRAK4 was highly expressed in Macrophages, Malignant and T-cells (Figure S2, Figures 5B and 5C). “
Figure 6. We highlight the prognostic impact of the expression of 4 genes that exhibited high levels of expression in adult macrophages.
Figures 7-10. The impact on the calculated HR for TRIM5 and MALT1 using 101 updated GBM patients was marginal and showed similar responses to the original 106 patients.
The results and discussion sections were substantially revised to reflect the impact of the expanded set of reactomes and genes impacted by TGFB2 methylation.
It may also be interesting to compare these findings on glioblastomas with other tumor types not typically associated with low MGMT levels. For instance, a subset of breast cancers exhibits similarly low MGMT expression (DOI: 10.1002/ijc.2910610308).
We performed a similar multivariate analysis for 778 TCGA breast cancer patients investigating TGFB1/2/3 and MGMT methylations controlling for age, breast cancer subtype, chemotherapy versus others, and interaction of TGFB2 methylation with Age. Trends for gene methylation did not achieve statistical significance for further investigation.
Response 5: There was not a significant decrease in HR for High.TGFB2meth group of patients (N=195 patients; HR (95% CI range) = 0.19 (0-7.98); P = 0.387)
There was not a significant increase in HR for High.TGFB1meth group of patients (N=194; HR (95% CI range) = 1.4 (0.77-2.55); P = 0.268)
There was not a significant decrease in HR for High.TGFB3meth group of patients (N=194; HR (95% CI range) = 0.76 (0.39-1.48); P = 0.416)
There was not a significant decrease in HR for High.MGMTmeth group of patients (N=194; HR (95% CI range) = 0.53 (0.26-1.05); P = 0.069)
There was not a significant increase in HR for Chemo_vs_NotChemo group of patients (N=99 chemo only treated; HR (95% CI range) = 1.65 (0.74-3.7); P = 0.221)
There was a significant increase in HR for AGE group of patients (N=778; HR (95% CI range) = 1.03 (1-1.05); P = 0.025)
There was not a significant decrease in HR for BRCA_Basal group of patients (N=120; HR (95% CI range) = 0.59 (0.22-1.55); P = 0.284)
There was not a significant decrease in HR for BRCA_Her2 group of patients (N=50; HR (95% CI range) = 0.24 (0.05-1.12); P = 0.069)
There was a significant decrease in HR for BRCA_LumA group of patients (N=365; HR (95% CI range) = 0.34 (0.16-0.73); P = 0.006)
There was not a significant decrease in HR for BRCA_LumB group of patients (N=142HR (95% CI range) = 0.7 (0.31-1.6); P = 0.401)
There was not a significant decrease in HR for BRCA_Normal group of patients (N=23; HR (95% CI range) = 0.86 (0.23-3.24); P = 0.819)
There was not a significant increase in HR for TGFB2meth by AGE interaction (HR (95% CI range) = 1.02 (0.96-1.08); P = 0.506)
Round 2
Reviewer 2 Report
Comments and Suggestions for Authors
The authors addressed the reviewers' concern and removed 5 patients with IDH-mutant brain tumors from their study without affecting major results. Therefore the manuscript improved significantly. There might be a conflict of interest in the Acknowledgment which should be solved with the editor: "V.T. as Chairman/CEO of Oncotelic Therapeutics" - how can the study not be independent from the company?
Author Response
Thank you for your comments. Your suggestions and careful considerations improved the quality of the manuscript.